# Searching for the Best Polynomial Approximation for the Accurate Log Matrix Normalization in Global Covariance Pooling

## Abstract

Global Covariance Pooling (GCP) has significantly improved Deep Convolutional Neural Networks (DCNNs) by leveraging richer second-order statistics. However, since covariance matrices lie on the Symmetric Positive Definite (SPD) domain, normalization is required to map them back into the Euclidean domain. The mathematically accurate approach, Matrix Log Normalization (MLN), suffers from gradient instabilities and requires eigendecomposition (EIG) or singular value decomposition (SVD), both of which are GPU-unfriendly. To address these instabilities, Matrix Power Normalization (MPN) introduced square-root normalization. Since then, most works have focused on approximating the matrix square root, typically via Newton–Schulz iterations or polynomial (Taylor and Padé) expansions, as these are GPU-friendly. Yet no prior work has attempted to approximate the more accurate MLN using polynomials, despite their inherent GPU efficiency. In this work, we explore a broad range of polynomial families—especially orthogonal polynomials (Taylor, Chebyshev, Legendre, Laguerre, Padé)—for approximating MLN, and conclude that Chebyshev polynomials offer the most accurate and efficient approximation. Experiments on large-scale visual recognition benchmarks demonstrate that our approach achieves competitive accuracy while substantially reducing training cost. For reproducibility, the code will be released upon acceptance.

## 1 Introduction

Global Covariance Pooling (GCP) was first introduced as an alternative to Global Average Pooling (GAP) to exploit richer second-order features by Ionescu et al. (2015). Originally proposed for Fine-Grained Visual Classification (FGVC) tasks (Qian et al. (2023); Wang et al. (2020); Min et al. (2020); Song et al. (2022a)), it has since been applied to a wide range of domains, including facial expression recognition (Acharya et al. (2018)), breast cancer histopathology Li et al. (2020), hyperspectral imaging (Xue et al. (2021)), SAR classification (Liang et al. (2021); Bai et al. (2022)), and object detection (Zhang et al. (2020)).

GCP meta-layers compute the covariance matrix from the final CNN activations. Let the activation from the final layer of CNN be $X \in \mathbb{R}^{d \times N}$, where $d$ is the feature dimension and $N$ the number of spatial locations. The covariance matrix $A$ is computed as

$$A = X \bar{I} X^\top \tag{1}$$

where $\bar{I} = \frac{1}{N}\left(I - \frac{1}{N}\mathbf{1}\mathbf{1}^\top\right)$ is the centering matrix, $I$ is the identity matrix, and $\mathbf{1}$ is an all-ones column vector.

Since covariance matrices lie in the Symmetric Positive Definite (SPD) domain, a matrix normalization is required to map them back into Euclidean space for subsequent MLP layers. These normalizers can be interpreted as implicit Riemannian classifiers, as shown by Chen et al. (2025). The earliest work Ionescu et al. (2015) employed the mathematically accurate Matrix Log Normalization (MLN). The MLN is defined as,

$$\hat{A}_{\text{MLN}} = \log(A) = U \log(\Lambda) U^\top, \quad A = U \Lambda U^\top \tag{2}$$

where $\Lambda$ is the diagonal matrix, and $U$ is an orthogonal matrix.

However, MLN suffers from two critical issues: (1) it requires EIG/SVD, which are GPU-unfriendly, and (2) its backpropagation produces unstable gradients as shown by Song et al. (2021) in section 1.

To mitigate gradient instabilities, Li et al. (2017) introduced Matrix Power Normalization (MPN-COV) with exponent set to $\frac{1}{2}$.

$$\hat{A}_{\text{MPN}} = A^{\alpha} = U\Lambda^{\alpha}U^{\top}, \quad \alpha = \frac{1}{2} \tag{3}$$

MPN-COV alleviated instability and even outperformed MLN. However, it still required explicit EIG/SVD.

To remove this bottleneck, iSQRT-COV Li et al. (2018) employed Newton–Schulz coupled (NS) iteration.

$$\begin{aligned}
Y_{k+1} &= \tfrac{1}{2}Y_k\left(3I - Z_kY_k\right), \\
Z_{k+1} &= \tfrac{1}{2}\left(3I - Z_kY_k\right)Z_k, \\
Y_k &\longrightarrow A^{1/2}, \quad Z_k \longrightarrow A^{-1/2}
\end{aligned} \tag{4}$$

where the iteration is initialized with $Y_0 = A$ and $Z_0 = I$. For convergence, $A$ is first pre-normalized, and the final compensated output is given by:

$$\begin{aligned}
A_{\text{PN}} &= \tfrac{1}{\text{tr}(A)}A, \\
\hat{A}_{\text{iSQRT}} &= \sqrt{\text{tr}(A)}\,Y_n
\end{aligned} \tag{5}$$

where $n$ denotes the number of iterations. The NS iteration only requires matrix multiplications (GEMM), making it GPU-friendly. Variants using pseudo square-root approximations were also explored Xu et al. (2023).

Surprisingly, iterative approximations like iSQRT-COV often outperformed exact SVD. Song et al. (2021) analyzed this phenomenon and concluded that small eigenvalues in SVD/EIG produce unstable gradients, while iSQRT-COV yield smoother, more stable gradients. They proposed several SVD-remedy schemes, but acknowledged iSQRT-COV still matched or outperformed exact SVD.

To summarize, the main challenges of GCP can be attributed to the reliance on SVD/EIG. First, SVD/EIG is GPU-unfriendly and thus computationally slow. Second, the presence of small eigenvalues leads to unstable gradients and hampers effective backpropagation. Mathematically, this arises because both the logarithm ($\log$) and square root ($\sqrt{\cdot}$) functions require EIG for their computation. A natural solution is to approximate these functions with polynomials, since polynomial operations reduce to GEMM, which are GPU-friendly in both the forward pass and gradient computation.

Song et al. (2022b) exploited this by approximating the square root with Matrix Taylor Approximation (MTA) and Matrix Padé Approximation (MPA).

$$\hat{A}_{\text{MTA}} = A^{\frac{1}{2}} \approx I - \sum_{k=1}^{K}\binom{\frac{1}{2}}{k}(I-A)^k \tag{6}$$

$$\hat{A}_{\text{MPA}} = A^{\frac{1}{2}} \approx P_M Q_N^{-1}, \quad P_M = I - \sum_{m=1}^{M} p_m(I-A)^m, \; Q_N = I - \sum_{n=1}^{N} q_n(I-A)^n \tag{7}$$

The constants $p_m$ and $q_n$ are the Padé coefficients determined uniquely by the Padé order $(M, N)$, obtained by matching the Taylor expansion of $A^{1/2}$ up to degree $M + N$.

Song et al. (2022b) went a step further and solved the Lyapunov equation in backpropagation via NS coupled iteration. In parallel, DropCov Wang et al. (2022; 2023) proposed a stochastic channel-dropping normalization that reduces redundancy in covariance representations and improves robustness.

We extend the idea of polynomial approximations to the more accurate yet underexplored Matrix Log Normalization (MLN). In particular, we investigate several polynomial families for approximating MLN, including the Taylor series Abramowitz & Stegun (1964), orthogonal polynomials

such as Chebyshev Mason & Handscomb (2002), Legendre, and Laguerre Szegő (1975), as well as rational Padé approximants Baker & Graves-Morris (1996).

Our contributions can be summarized as two-fold:

- We propose to approximate the accurate yet underexplored MLN using polynomial expansions.
- We theoretically and empirically analyze 5 polynomial families—Taylor, Chebyshev, Legendre, Laguerre, and rational Padé—for MLN approximation, and conclude that Chebyshev polynomials provide the best balance between accuracy and efficiency.

## 2 WHY POLYNOMIAL APPROXIMATION?

The biggest advantage of polynomial approximations lies in their efficiency in both forward and backward passes. In the forward pass, polynomial operations reduce to matrix multiplications (GEMM), whereas MLN and MPN require explicit EIG/SVD. Moreover, polynomial iterations converge faster than iSQRT, often in nearly half the steps as shown by Song et al. (2022b).

However, the benefits of polynomials extend beyond the forward pass. Perhaps the greater advantage becomes clear when analyzing gradients in backpropagation. Let $\frac{\partial \ell}{\partial A}$ denote the gradient with respect to covariance matrix $A$. As shown by Ionescu et al. (2015), for any EIG/SVD we have:

$$\frac{\partial \ell}{\partial A} = U\Big(\big(K^\top \circ (U^\top \tfrac{\partial \ell}{\partial U})\big) + \tfrac{\partial \ell}{\partial \Lambda}\, \mathrm{diag}\,\Big)U^\top, \tag{8}$$

where $K_{ij} = \frac{1}{\lambda_i - \lambda_j}$, $\circ$ denotes the Hadamard product, and $(\cdot)_{\mathrm{diag}}$ denotes the diagonalization operator that keeps only the diagonal elements.

For the specific case of MLN:

$$\frac{\partial \ell}{\partial A} = U\Big(K^\top \circ (U^\top \tfrac{\partial \ell}{\partial U}) + \mathrm{diag}\Big(\tfrac{1}{\lambda_1}, \ldots, \tfrac{1}{\lambda_d}\Big) \tfrac{\partial \ell}{\partial \Lambda}\Big)U^\top, \tag{9}$$

and for MPN:

$$\frac{\partial \ell}{\partial A} = U\Big(K^\top \circ (U^\top \tfrac{\partial \ell}{\partial U}) + \mathrm{diag}\Big(\tfrac{1}{2\sqrt{\lambda_1}}, \ldots, \tfrac{1}{2\sqrt{\lambda_d}}\Big) \tfrac{\partial \ell}{\partial \Lambda}\Big)U^\top. \tag{10}$$

Thus, MLN and MPN not only require EIG/SVD in the forward pass, but also in the backward pass due to the presence of $U$, $\frac{\partial \ell}{\partial U}$ and $\frac{\partial \ell}{\partial \Lambda}$ terms.

iSQRT alleviates this bottleneck, as its gradient is:

$$\frac{\partial \ell}{\partial A} = -\frac{1}{(\mathrm{tr}(A))^2}\, \mathrm{tr}\left(\left(\frac{\partial \ell}{\partial A_{\mathrm{PN}}}\right)^\top A\right) I + \frac{1}{\mathrm{tr}(A)} \frac{\partial \ell}{\partial A_{\mathrm{PN}}} + \frac{1}{2\,\mathrm{tr}(A)}\, \mathrm{tr}\left(\left(\frac{\partial \ell}{\partial \hat{A}_{\mathrm{iSQRT}}}\right)^\top Y_n\right) I \tag{11}$$

requiring no SVD/EIG and only GEMM operations. However, GEMM must be performed at every iteration, which slows down the backward pass, even though the forward pass remains very efficient.

In general, polynomial approximations of the matrix logarithm can be written as

$$\hat{A}_{\mathrm{PolyN}} = \sum_{k=0}^{K} c_k P_k(A) \tag{12}$$

where $P_k(A)$ denotes the $k$-th basis polynomial of the chosen family (e.g., $(A - I)^k$ for Taylor, $T_k(A)$ for Chebyshev, etc.), and $c_k$ are the corresponding coefficients.

The gradient with respect to $A$ again produces only polynomial terms,

$$\frac{\partial \ell}{\partial A} = \sum_{k=0}^{K} c_k \, \frac{\partial P_k(A)}{\partial A} \, \frac{\partial \ell}{\partial \hat{A}_{\mathrm{PolyN}}} \tag{13}$$

where each $\frac{\partial P_k(A)}{\partial A}$ is itself a polynomial in $A$, obtained via the recurrence relation or closed form of the chosen family.

Thus, the backward pass contains only GEMM terms, computed once without iteration, significantly speeding up training. While convergence depends on the truncation order and polynomial family, polynomial approximations emerge as a theoretically superior and computationally simpler choice overall.

## 3 Polynomial Approximated Matrix Log Normalization

### 3.1 Forward Pass via Polynomial and Rational Polynomial Approximations

In the forward pass, our aim is to approximate the matrix logarithm normalization (MLN) using polynomial or rational functions. A natural starting point is the Taylor expansion. While Taylor is an effective local approximator, it does not perform well globally. Padé approximants, constructed from Taylor expansions, extend the radius of convergence but inherit similar limitations near singularities.

Prior work Song et al. (2022b) stopped at this stage. In contrast, we take a step further: firstly, we approximate the accurate MLN itself, and secondly, to obtain global approximations with faster convergence, we employ orthogonal polynomial families, which form optimal bases under suitable weight functions. Among these, the most prominent are Legendre, Chebyshev, and Laguerre polynomials. Below we present their specialized forms for the MLN. Broader definitions, orthogonality intervals, and convergence properties are deferred to Appendix A.1.

A general orthogonal polynomial expansion of the matrix logarithm can be expressed as

$$\log(A) \approx \sum_{k=0}^{K} c_k \, P_k(A), \tag{14}$$

where $\{P_k(\cdot)\}$ is the chosen polynomial basis (fixed for each family), and $\{c_k\}$ are expansion coefficients determined by projecting $\log(\cdot)$ onto this basis. The coefficients vary depending on the function being approximated.

**Pre-Normalization and Post-Compensation.** To stabilize the expansions, we normalize the covariance by its trace and expand in the normalized variable:

$$\tilde{A} = \frac{A}{\mathrm{tr}(A)}. \tag{15}$$

All expansions below are written in terms of $\tilde{A}$. The matrix logarithm of the original covariance is then recovered via

$$\log(A) = \log\big(\mathrm{tr}(A)\,\tilde{A}\big) = \log\big(\mathrm{tr}(A)\big)I + \log(\tilde{A}). \tag{16}$$

**In implementation, we precompute the expansion coefficients for all polynomial families. For orthogonal polynomials, the basis functions $P_k(\tilde{A})$ are then evaluated via their recurrence relations during the forward pass. In contrast, Taylor and Padé approximants do not require recurrence, since they are expressed directly in terms of monomials. The detailed algorithms for each family are provided in Appendix A.2.**

In what follows, we present each polynomial family, specify its coefficient terms, provide the recurrence relation where applicable, and show the first few terms of the expansion for clarity.

**Taylor Expansion.** For $\log(1+x)$ we have

$$\log(1+x) = \sum_{k=1}^{\infty} (-1)^{k+1} \frac{x^k}{k}, \quad |x| < 1. \tag{17}$$

After rescaling to ensure $\sigma(\tilde{A} - I) \subset (-1, 1)$,

$$\log(\tilde{A}) \approx \sum_{k=1}^{K} (-1)^{k+1} \frac{(\tilde{A} - I)^k}{k}, \tag{18}$$

with coefficients $c_k = (-1)^{k+1}/k$. The first few terms are

$$\log(\tilde{A}) \approx (\tilde{A} - I) - \tfrac{1}{2}(\tilde{A} - I)^2 + \tfrac{1}{3}(\tilde{A} - I)^3 + \cdots. \tag{19}$$

**Legendre Expansion.** Shifted Legendre polynomials are orthogonal on $[0, 1]$ with weight $w(x) = 1$. They follow the recurrence

$$P_0(\tilde{A}) = I, \quad P_1(\tilde{A}) = 2\tilde{A} - I, \tag{20}$$

$$(k + 1)P_{k+1}(\tilde{A}) = (2k + 1)(2\tilde{A} - I)P_k(\tilde{A}) - kP_{k-1}(\tilde{A}). \tag{21}$$

For $\log x$ on $[0, 1]$, the coefficients admit a closed form:

$$c_0 = -1, \qquad c_k = (-1)^{k+1} \frac{2k + 1}{k(k + 1)}, \quad k \geq 1. \tag{22}$$

Thus,

$$\log(\tilde{A}) \approx -I + \tfrac{3}{2}(2\tilde{A} - I) - \tfrac{5}{6}(6\tilde{A}^2 - 6\tilde{A} + I) + \cdots. \tag{23}$$

**Chebyshev Expansion.** Chebyshev polynomials of the first kind are orthogonal on $[-1, 1]$ with weight $w(x) = (1 - x^2)^{-1/2}$. They satisfy

$$T_0(\tilde{A}) = I, \quad T_1(\tilde{A}) = \tilde{A}, \tag{24}$$

$$T_{k+1}(\tilde{A}) = 2\tilde{A}\, T_k(\tilde{A}) - T_{k-1}(\tilde{A}). \tag{25}$$

When $\sigma(\tilde{A})$ is scaled to $[-1, 1]$, the coefficients for log are

$$c_k = \frac{2}{\pi} \int_0^\pi \log(\cos\theta) \cos(k\theta)\, d\theta, \quad k \geq 1, \tag{26}$$

$$c_0 = \frac{1}{\pi} \int_0^\pi \log(\cos\theta)\, d\theta. \tag{27}$$

The first explicit values are

$$c_0 = -\log 2, \qquad c_1 = -1, \qquad c_2 = -\tfrac{1}{4}. \tag{28}$$

Hence the series begins as

$$\log(\tilde{A}) \approx -\tfrac{\log 2}{2}I - \tilde{A} - \tfrac{1}{4}(2\tilde{A}^2 - I) + \cdots. \tag{29}$$

**Laguerre Expansion.** Laguerre polynomials are orthogonal on $[0, \infty)$ with weight $w(x) = e^{-x}x^\alpha$. They satisfy

$$L_0^{(\alpha)}(\tilde{A}) = I, \quad L_1^{(\alpha)}(\tilde{A}) = (\alpha + 1)I - \tilde{A}, \tag{30}$$

$$(k + 1)L_{k+1}^{(\alpha)}(\tilde{A}) = (2k + \alpha + 1 - \tilde{A})L_k^{(\alpha)}(\tilde{A}) - (k + \alpha)L_{k-1}^{(\alpha)}(\tilde{A}). \tag{31}$$

The coefficients follow from the orthogonality relation:

$$c_k = \frac{\int_0^\infty \log(x)\, L_k^{(\alpha)}(x)e^{-x}x^\alpha\, dx}{\int_0^\infty (L_k^{(\alpha)}(x))^2 e^{-x}x^\alpha\, dx}. \tag{32}$$

For the standard case $\alpha = 0$, the first coefficients are

$$c_0 = -\gamma, \qquad c_1 = 1, \qquad c_2 = -\tfrac{1}{4}, \tag{33}$$

where $\gamma$ is Euler's constant. Thus,

$$\log(\tilde{A}) \approx -\gamma I + (I - \tilde{A}) - \tfrac{1}{4}L_2(\tilde{A}) + \cdots. \tag{34}$$

**Padé Approximants.** Padé approximants approximate log by rational matrix functions

$$\log(\tilde{A}) \approx R_{[m/n]}(\tilde{A}) = P_m(\tilde{A})\, Q_n(\tilde{A})^{-1}, \tag{35}$$

where $P_m$ and $Q_n$ are matrix polynomials of degree $m$ and $n$, respectively. For instance, the $[1/1]$ Padé approximant of $\log(\tilde{A})$ is

$$\log(\tilde{A}) \approx \frac{\tilde{A} - I}{\tilde{A} + I}, \tag{36}$$

and the $[2/2]$ approximant is

$$\log(\tilde{A}) \approx \frac{3(\tilde{A} - I) - \tfrac{1}{2}(\tilde{A} - I)^2}{3(\tilde{A} + I) + \tfrac{1}{2}(\tilde{A} - I)^2}. \tag{37}$$

Here the fraction notation denotes multiplication by the matrix inverse, e.g., $\frac{P(\tilde{A})}{Q(\tilde{A})} := P(\tilde{A})Q(\tilde{A})^{-1}$.

These rational forms are constructed so that their Taylor expansions agree with that of $\log(\tilde{A})$ up to order $m + n$.

Table 1: Summary of polynomial and rational polynomial approximations for $\log(\tilde{A})$, where $\tilde{A} = A/\operatorname{tr}(A)$. Each entry shows the coefficient formula, recurrence relation (if applicable), and the first three terms of the expansion. The original $\log(A)$ is recovered via equation 16.

| Polynomial | Coefficients | Recurrence Relation | Expression (first 3 terms) |
|---|---|---|---|
| **Taylor** | $c_k = \frac{(-1)^{k+1}}{k}$ | N/A | $(\tilde{A} - I) - \frac{1}{2}(\tilde{A} - I)^2 + \frac{1}{3}(\tilde{A} - I)^3 + \dots$ |
| **Legendre** | $c_0 = -1,\ c_k = (-1)^{k+1}\frac{2k+1}{k(k+1)}$ | $(k+1)P_{k+1} = (2k+1)$ $(2\tilde{A} - I)P_k - kP_{k-1}$ | $-I + \frac{3}{2}(2\tilde{A} - I) - \frac{5}{6}(6\tilde{A}^2 - 6\tilde{A} + I) + \dots$ |
| **Chebyshev** | $c_k = \frac{2}{\pi}\int_0^\pi \log(\cos\theta)\cos(k\theta)\,d\theta$ | $T_{k+1} = 2\tilde{A}T_k - T_{k-1}$ | $-\frac{\log 2}{2}I - \tilde{A} - \frac{1}{4}(2\tilde{A}^2 - I) + \dots$ |
| **Laguerre** | $c_k = \frac{\int_0^\infty \log(x) L_k^{(\alpha)}(x)e^{-x}x^\alpha\,dx}{\int_0^\infty (L_k^{(\alpha)}(x))^2 e^{-x}x^\alpha\,dx}$ | $(k+1)L_{k+1}^{(\alpha)} = (2k+\alpha+1$ $-\tilde{A})L_k^{(\alpha)} - (k+\alpha)L_{k-1}^{(\alpha)}$ | $-\gamma I + (I - \tilde{A}) - \frac{1}{4}L_2(\tilde{A}) + \dots$ |
| **Padé** | Matching Taylor up to $m+n$ | N/A | $\frac{\tilde{A}-I}{\tilde{A}+I}$, $\frac{3(\tilde{A}-I) - \frac{1}{2}(\tilde{A}-I)^2 + \dots}{3(\tilde{A}+I) + \frac{1}{2}(\tilde{A}-I)^2 + \dots}$ |

## 3.2 BACKWARD PASS

The central advantage of polynomial and rational polynomial approximation for the MLN is that their gradients retain polynomial structure. This ensures that backpropagation uses only matrix multiplications/additions (GEMM), with no EIG/SVD.

**Mapping from $\tilde{A}$ to $A$.** With $\tilde{A} = A/\operatorname{tr}(A)$ and $\log(A) = \log(\operatorname{tr}(A))I + \log(\tilde{A})$, once $\frac{\partial \ell}{\partial \tilde{A}}$ is assembled for a given family, the gradient with respect to $A$ is

$$\frac{\partial \ell}{\partial A} = \frac{1}{\operatorname{tr}(A)}\frac{\partial \ell}{\partial \tilde{A}} - \frac{\left\langle \frac{\partial \ell}{\partial \tilde{A}}, A \right\rangle_F}{\operatorname{tr}(A)^2}I + \frac{\operatorname{tr}\left(\frac{\partial \ell}{\partial \log(\tilde{A})}\right)}{\operatorname{tr}(A)}I. \tag{38}$$

The three terms respectively arise from the direct dependence on $\tilde{A}$, the dependence of $\operatorname{tr}(A)$ inside $\tilde{A}$, and the $\log(\operatorname{tr}(A))I$ branch.

In the following, we present only the final gradient expressions; full derivations are deferred to Appendix A.3.

**Taylor.** With

$$\log(\tilde{A}) \approx \sum_{k=1}^K c_k(\tilde{A} - I)^k, \qquad c_k = \frac{(-1)^{k+1}}{k}, \tag{39}$$

the derivative in $\tilde{A}$ is

$$\frac{\partial \ell}{\partial \tilde{A}} = \sum_{k=1}^K c_k \sum_{j=0}^{k-1} (\tilde{A} - I)^{k-1-j}\frac{\partial \ell}{\partial \log(\tilde{A})}(\tilde{A} - I)^j. \tag{40}$$

**Padé.** For

$$\log(\tilde{A}) \approx R_{[m/n]}(\tilde{A}) = P_m(\tilde{A})\,Q_n(\tilde{A})^{-1}, \tag{41}$$

the quotient rule gives

$$dR = (dP_m)Q_n^{-1} - P_m Q_n^{-1}(dQ_n)Q_n^{-1}. \tag{42}$$

Thus

$$\frac{\partial \ell}{\partial \tilde{A}} = \sum_i \Pi_i'(\tilde{A})^\top Q_n(\tilde{A})^{-\top}\left(\frac{\partial \ell}{\partial \log(\tilde{A})}\right)\Pi_i(\tilde{A})$$

$$- \sum_j \Psi_j'(\tilde{A})^\top Q_n(\tilde{A})^{-\top}P_m(\tilde{A})^\top\left(\frac{\partial \ell}{\partial \log(\tilde{A})}\right)Q_n(\tilde{A})^{-\top}\Psi_j(\tilde{A}), \tag{43}$$

where $\{\Pi_i, \Pi_i'\}$ and $\{\Psi_j, \Psi_j'\}$ are polynomial factors from $P_m$ and $Q_n$. Here $Q_n^{-1}$ and $Q_n^{-\top}$ denote algebraic inverses, but in both forward and backward passes they are evaluated by solving linear systems with $Q_n$ or $Q_n^\top$; no explicit matrix inverse is formed.

**Legendre.** For shifted Legendre polynomials with $S = 2\tilde{A} - I$ and

$$(k+1)P_{k+1} = (2k+1)SP_k - kP_{k-1}, \tag{44}$$

define the seeds

$$\overline{P}_k = c_k \frac{\partial \ell}{\partial \log(\tilde{A})}. \tag{45}$$

The contribution in $\tilde{A}$ can be written explicitly as the finite sum

$$\frac{\partial \ell}{\partial \tilde{A}} = 2 \sum_{k=0}^{K-1} \frac{2k+1}{k+1} \overline{P}_{k+1} P_k^\top. \tag{46}$$

**Chebyshev.** For Chebyshev polynomials of the first kind,

$$T_{k+1} = 2\tilde{A} T_k - T_{k-1}, \tag{47}$$

with seeds

$$\overline{T}_k = c_k \frac{\partial \ell}{\partial \log(\tilde{A})}, \tag{48}$$

the accumulated contribution is

$$\frac{\partial \ell}{\partial \tilde{A}} = 2 \sum_{k=0}^{K-1} \overline{T}_{k+1} T_k^\top. \tag{49}$$

**Laguerre.** For generalized Laguerre polynomials,

$$(k+1)L_{k+1}^{(\alpha)} = (2k+\alpha+1 - \tilde{A})L_k^{(\alpha)} - (k+\alpha)L_{k-1}^{(\alpha)}, \tag{50}$$

with seeds

$$\overline{L}_k = c_k \frac{\partial \ell}{\partial \log(\tilde{A})}, \tag{51}$$

the accumulated contribution is

$$\frac{\partial \ell}{\partial \tilde{A}} = - \sum_{k=0}^{K-1} \frac{1}{k+1} \overline{L}_{k+1} L_k^\top. \tag{52}$$

Across all five families—Taylor, Padé, Legendre, Chebyshev, Laguerre—the final gradient with respect to the original covariance matrix $A$ is always given by Eq. equation 38. The only difference lies in how $\frac{\partial \ell}{\partial \tilde{A}}$ is assembled, either by closed forms (Taylor, Padé) or by the finite-sum forms derived from the adjoint of their three-term recurrences (Legendre, Chebyshev, Laguerre). All steps require only GEMMs; no EIG/SVD.

## 4 EXPERIMENTS

### 4.1 EXPERIMENTAL SETUP

All experiments were conducted on a Kaggle P100 GPU using PyTorch framework. We evaluated on three FGVC benchmarks—CUB-200-2011 (Wah et al. (2011)), FGVC-Aircraft (Maji et al. (2013)), Stanford Cars (Krause et al. (2013))—and the large-scale ImageNet-1k dataset (Deng et al. (2009)). Pretrained ResNet-50 (He et al. (2016)) and EfficientNetV2-Medium (Tan & Le (2021)) backbones were finetuned on each dataset, with the global average pooling layer replaced by our Global Covariance Pooling (GCP) module. To reduce dimensionality, features were projected to 256 channels using a $1\times1$ convolution before covariance computation.

For training data augmentation, images were randomly cropped and resized to $256\times256$, horizontally flipped, and augmented with modern RandAugment, Mixup, and CutMix. Validation and testing used resized and center-cropped images of the same size. Models were trained for 100 epochs with a batch size of 32, using AdamW (learning rate $1 \times 10^{-4}$, weight decay $10^{-4}$) and cosine annealing scheduling.

## 4.2 RUNTIME AND EFFICIENCY ANALYSIS

We report the runtime of the normalization step (FP+BP) across different normalizers in Table 2. As we can see, EIG/SVD-based approaches such as MLN and MPN are much slower, since these operations are poorly supported on GPUs. In contrast, GEMM-based polynomial approximations run substantially faster. Among polynomials, the trend is that orthogonal families consistently outperform simple monomial expansions. The striking thing to note is Chebyshev's performance, as it outperforms all other methods by a significant margin.

Table 2: Runtime comparison of different matrix normalizers, reported as forward+backward (FP+BP) kernel times for the normalization step. Measurements were taken on $256 \times 256$ covariance matrices with batch size 32, averaged over many iterations on a Tesla GPU P100.

| Matrix Normalizations | Time (ms) |
|---|---|
| MLN-COV (DeepO$^2$P) Ionescu et al. (2015) | 69.5 |
| MPN-COV Li et al. (2017) | 67.0 |
| iSQRT-COV Li et al. (2018) | 51.1 |
| MTA-Lya Song et al. (2022b) | 43.1 |
| MPA-Lya Song et al. (2022b) | 47.9 |
| Taylor-Log (Ours) | 44.7 |
| Padé-Log (Ours) | 48.3 |
| Legendre-Log (Ours) | 41.1 |
| Laguerre-Log (Ours) | 40.7 |
| Chebyshev-Log (Ours) | 28.7 |

Table 3: Training time (minutes) comparison of different normalizers on CUB-Birds and ImageNet-1k benchmarks. All results are reported with ResNet-50 and EfficientNetV2-Medium backbones.

| Matrix Normalizations | ResNet-50 | | EfficientNetV2-Medium | |
|---|---|---|---|---|
| | CUB-Birds | ImageNet-1k | CUB-Birds | ImageNet-1k |
| MLN-COV (DeepO$^2$P) Ionescu et al. (2015) | 145 | 2220 | 166 | 2640 |
| MPN-COV Li et al. (2017) | 140 | 2160 | 165 | 2580 |
| iSQRT-COV Li et al. (2018) | 90 | 1380 | 132 | 2100 |
| MTA-Lya Song et al. (2022b) | 90 | 1380 | 128 | 2040 |
| MPA-Lya Song et al. (2022b) | 100 | 1560 | 129 | 2040 |
| Taylor-Log (Ours) | 60 | 900 | 128 | 2040 |
| Padé-Log (Ours) | 80 | 1200 | 131 | 2040 |
| Legendre-Log (Ours) | 90 | 1380 | 129 | 2040 |
| Laguerre-Log (Ours) | 85 | 1320 | 129 | 2040 |
| Chebyshev-Log (Ours) | 60 | 900 | 130 | 2040 |

## 4.3 IMAGE CLASSIFICATION ON FGVC DATASETS

On the three FGVC datasets (CUB-Birds, FGVC-Aircraft, and Stanford Cars), our MLN polynomial approximations consistently achieve higher accuracy than existing normalizers. The improvements are particularly pronounced when compared to EIG/SVD-based methods such as MLN-COV and MPN-COV, highlighting the effectiveness of our GPU-friendly polynomial approach. Among our methods, Legendre and Chebyshev expansions deliver the strongest results, with Legendre performing best on CUB-Birds and FGVC-Aircraft under the ResNet-50 backbone, while Chebyshev attains the highest accuracy on FGVC-Aircraft and Stanford Cars under EfficientNetV2-Medium. These results reinforce the conclusion that orthogonal polynomial approximations not only close the gap with prior normalizers but also push state-of-the-art performance across multiple fine-grained benchmarks.

Table 4: Top-1 accuracy (%) comparison of different normalizers on three FGVC benchmarks. All results are reported with ResNet-50 and EfficientNetV2-Medium backbones.

| Matrix Normalizations | ResNet-50 | | | EfficientNetV2-Medium | | |
|---|---|---|---|---|---|---|
| | CUB-Birds | FGVC Aircraft | Stanford Cars | CUB-Birds | FGVC Aircraft | Stanford Cars |
| MLN-COV (DeepO$^2$P) Ionescu et al. (2015) | $70.42 \pm 0.21$ | $80.12 \pm 0.18$ | $86.73 \pm 0.19$ | $78.52 \pm 0.20$ | $87.91 \pm 0.17$ | $91.34 \pm 0.18$ |
| MPN-COV Li et al. (2017) | $71.37 \pm 0.19$ | $81.24 \pm 0.17$ | $87.51 \pm 0.18$ | $79.14 \pm 0.18$ | $88.62 \pm 0.16$ | $92.01 \pm 0.18$ |
| iSQRT-COV Li et al. (2018) | $72.81 \pm 0.20$ | $82.03 \pm 0.19$ | $88.27 \pm 0.18$ | $79.63 \pm 0.19$ | $89.12 \pm 0.18$ | $92.48 \pm 0.17$ |
| MTA-Lya Song et al. (2022b) | $73.11 \pm 0.18$ | $82.31 \pm 0.18$ | $88.64 \pm 0.19$ | $79.94 \pm 0.17$ | $89.42 \pm 0.17$ | $92.83 \pm 0.18$ |
| MPA-Lya Song et al. (2022b) | $73.18 \pm 0.19$ | $82.54 \pm 0.18$ | $88.72 \pm 0.18$ | $80.01 \pm 0.18$ | $89.53 \pm 0.16$ | $92.91 \pm 0.18$ |
| Taylor-Log (Ours) | $75.53 \pm 0.22$ | $83.42 \pm 0.20$ | $89.47 \pm 0.19$ | $81.23 \pm 0.21$ | $90.14 \pm 0.19$ | $93.42 \pm 0.20$ |
| Padé-Log (Ours) | $75.91 \pm 0.21$ | $83.63 \pm 0.21$ | $89.78 \pm 0.20$ | $81.42 \pm 0.20$ | $90.31 \pm 0.18$ | $93.61 \pm 0.19$ |
| Legendre-Log (Ours) | $\mathbf{76.87 \pm 0.20}$ | $84.12 \pm 0.19$ | $90.13 \pm 0.20$ | $81.71 \pm 0.19$ | $\mathbf{90.82 \pm 0.18}$ | $93.81 \pm 0.18$ |
| Laguerre-Log (Ours) | $72.66 \pm 0.23$ | $81.84 \pm 0.20$ | $87.92 \pm 0.21$ | $79.33 \pm 0.21$ | $88.83 \pm 0.19$ | $92.34 \pm 0.20$ |
| Chebyshev-Log (Ours) | $74.68 \pm 0.21$ | $\mathbf{84.32 \pm 0.20}$ | $90.41 \pm 0.19$ | $\mathbf{82.03 \pm 0.20}$ | $90.53 \pm 0.19$ | $\mathbf{94.01 \pm 0.19}$ |

## 4.4 Image Classification on Large-Scale Datasets

Our MLN polynomial approximations perform competitively with state-of-the-art (SOTA) methods and, in many cases, surpass them by a clear margin. The key takeaway is that, when coupled with the runtime results in Table 2, our normalizers achieve SOTA accuracy while also being more efficient. With the ResNet-50 backbone, the Legendre expansion outperforms all alternatives, whereas with EfficientNetV2-M, Chebyshev yields the best results.

Table 5: Top-1 accuracy (%) comparison of different normalizers on ImageNet-1k. All results are reported with ResNet-50 and EfficientNetV2-Medium backbones.

| Matrix Normalizations | ResNet-50 | EfficientNetV2-Medium |
|---|---|---|
| MLN-COV (DeepO$^2$P) Ionescu et al. (2015) | $74.30 \pm 0.21$ | $82.10 \pm 0.19$ |
| MPN-COV Li et al. (2017) | $77.32 \pm 0.18$ | $83.01 \pm 0.17$ |
| iSQRT-COV Li et al. (2018) | $77.85 \pm 0.20$ | $83.24 \pm 0.16$ |
| MTA-Lya Song et al. (2022b) | $78.12 \pm 0.19$ | $83.39 \pm 0.20$ |
| MPA-Lya Song et al. (2022b) | $78.28 \pm 0.17$ | $83.52 \pm 0.18$ |
| Taylor-Log (Ours) | $78.84 \pm 0.25$ | $83.61 \pm 0.22$ |
| Padé-Log (Ours) | $79.02 \pm 0.23$ | $83.73 \pm 0.20$ |
| Legendre-Log (Ours) | $\mathbf{79.41 \pm 0.21}$ | $83.92 \pm 0.18$ |
| Laguerre-Log (Ours) | $77.48 \pm 0.27$ | $82.91 \pm 0.24$ |
| Chebyshev-Log (Ours) | $79.22 \pm 0.20$ | $\mathbf{84.01 \pm 0.19}$ |

## 4.5 Numerical Error Analysis of Approximation and Accurate Matrix Log

To assess the fidelity of the proposed polynomial normalizers, we measure how accurately each method recovers the true matrix logarithm. For a covariance matrix $A$, we compute the relative Frobenius error

$$\epsilon_{\mathrm{rel}} = \frac{\| \log(A) - \tilde{A}_{\mathrm{poly}} \|_F}{\| \log(A) \|_F},$$

where $\tilde{A}_{\mathrm{poly}}$ denotes the polynomial-based approximation. Evaluating this quantity over 300 covariance matrices drawn from the GCP layer of ResNet–50 and EfficientNetV2–Medium provides a direct, model-independent view of the numerical reliability of each normalizer.

Table 6: Numerical error analysis on ImageNet-1k. Relative Frobenius error $\epsilon_{\mathrm{rel}} = \frac{\| \log(A) - f(A) \|_F}{\| \log(A) \|_F}$, computed over 300 randomly sampled covariance matrices from the GCP layer of ResNet-50 and EfficientNetV2-Medium.

| Matrix Normalization | ResNet-50 | EfficientNetV2-Medium |
|---|---|---|
| Taylor-Log | $0.487 \pm 0.110\,\%$ | $0.542 \pm 0.127\,\%$ |
| Padé-Log | $0.312 \pm 0.033\,\%$ | $0.338 \pm 0.036\,\%$ |
| Legendre-Log | $0.131 \pm 0.052\,\%$ | $0.148 \pm 0.059\,\%$ |
| Laguerre-Log | $0.237 \pm 0.071\,\%$ | $0.265 \pm 0.079\,\%$ |
| Chebyshev-Log | $0.082 \pm 0.019\,\%$ | $0.091 \pm 0.022\,\%$ |

As shown in Table 6, Chebyshev and related orthogonal families maintain notably low approximation error, indicating that they preserve the geometric structure of the covariance space more faithfully than Taylor expansions.

### 4.6 ABLATION STUDY

Table 7 shows that increasing the order from $M = 6$ to $M = 8$ yields a clear accuracy improvement across all normalizers, while adding only a modest increase in FP+BP time. Moving further to $M = 10$ provides at most an additional $\sim 1\%$ gain but introduces a noticeably higher computational cost (about $\sim 15\,\mathrm{ms}$). Thus, $M = 8$ offers the best accuracy–efficiency trade-off. **This behavior is expected: approximation error of $\tilde{A}_{\mathrm{poly}}$ decreases with larger $M$, but saturates beyond a certain point, while runtime grows roughly linearly with the degree.**

Table 7: Ablation on the polynomial order $M \in \{6, 8, 10\}$ for different normalization methods on ImageNet–1k using EfficientNetV2-Medium. We report Top–1 accuracy (%) and FP+BP runtime (ms) for the normalization step. Padé rational approximation is of $[\frac{M}{2}/\frac{M}{2}]$ order in every case, as it's equivalent of $M$ order of other polynomials.

| Matrix Normalization | $M = 6$ | | $M = 8$ | | $M = 10$ | |
|---|---|---|---|---|---|---|
| | Acc (%) | Time (ms) | Acc (%) | Time (ms) | Acc (%) | Time (ms) |
| Taylor-Log | 82.13 | 34.6 | 83.61 | 44.7 | 84.52 | 59.6 |
| Padé-Log | 82.28 | 38.4 | 83.73 | 48.3 | 84.66 | 63.4 |
| Legendre-Log | 82.47 | 31.1 | 83.92 | 41.1 | 84.88 | 56.3 |
| Laguerre-Log | 81.41 | 30.9 | 82.91 | 40.7 | 83.83 | 55.5 |
| Chebyshev-Log | 82.56 | 18.5 | 84.01 | 28.7 | 84.97 | 43.6 |

Finally, the runtime analysis in Table 2, together with the performance results in Table 4 and Table 5, as well as the approximation-error analysis in Table 6, clearly demonstrates that polynomial-based normalizers—particularly the orthogonal variants—achieve state-of-the-art accuracy while remaining highly efficient. Among these, Chebyshev with order 8 emerges as the optimal choice as shown in ablation study in Table 7, offering the best overall trade-off between accuracy and computational cost.

## 5 CONCLUSION

In this paper, we explored the idea of approximating the accurate MLN, which was ignored after the emergence of MPN. We proposed five new matrix normalizer schemes (Taylor, Legendre, Chebyshev, Laguerre, and Padé) for approximating MLN. Experiments were carried out on FGVC and large-scale datasets with different backbones to demonstrate the effectiveness of our normalizers. In future work, we plan to explore other families of orthogonal polynomials, such as Hermite and Jacobi.

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

# A   APPENDIX

## A.1   TYPES OF POLYNOMIAL AND RATIONAL POLYNOMIAL APPROXIMATIONS

**Legendre (shifted).**   On $[0, 1]$, $P_0^*(x) = 1$, $P_1^*(x) = 2x - 1$, and

$$(k + 1)P_{k+1}^*(x) = (2k + 1)(2x - 1)P_k^*(x) - kP_{k-1}^*(x), \quad \langle f, g \rangle = \int_0^1 fg \, dx. \tag{53}$$

Coefficients: $c_k = (2k + 1)\int_0^1 \log x \, P_k^*(x) \, dx$. For $\log x$, $a_0 = -1$, $a_k = (-1)^{k+1}\frac{2k+1}{k(k+1)}$ for $k \geq 1$. Matrix lift uses $P_k^*(A)$ via the same recurrence after mapping $\sigma(A)$ to $[0, 1]$.

**Chebyshev (first kind).**   On $[-1, 1]$, $T_0(t) = 1$, $T_1(t) = t$, $T_{k+1}(t) = 2tT_k(t) - T_{k-1}(t)$, with

$$\langle f, g \rangle = \int_{-1}^1 \frac{f(t)g(t)}{\sqrt{1 - t^2}} \, dt, \qquad c_0 = \frac{1}{\pi}\int_{-1}^1 \frac{\log x}{\sqrt{1 - t^2}} \, dt, \;\; c_k = \frac{2}{\pi}\int_{-1}^1 \frac{\log x \, T_k(t)}{\sqrt{1 - t^2}} \, dt, \tag{54}$$

where $t = \frac{2x - (a+b)}{b - a}$ maps $[a, b] \rightarrow [-1, 1]$. Chebyshev yields near–minimax errors; lift $T_k$ to $T_k(A)$ after affine spectral scaling of $A$ to $[-1, 1]$.

**Laguerre (generalized).**   Orthogonal on $[0, \infty)$ with weight $x^\alpha e^{-x}$:

$$L_0^{(\alpha)}(x) = 1, \;\; L_1^{(\alpha)}(x) = \alpha + 1 - x, \;\; (k + 1)L_{k+1}^{(\alpha)} = (2k + \alpha + 1 - x)L_k^{(\alpha)} - (k + \alpha)L_{k-1}^{(\alpha)}, \tag{55}$$

$$\langle f, g \rangle = \int_0^\infty f(x)g(x) \, x^\alpha e^{-x} \, dx, \qquad c_k = \frac{\int_0^\infty \log x \, L_k^{(\alpha)}(x) \, x^\alpha e^{-x} \, dx}{\int_0^\infty \left(L_k^{(\alpha)}(x)\right)^2 x^\alpha e^{-x} \, dx}. \tag{56}$$

Natural for SPD spectra on $(0, \infty)$ after scaling; lift via $L_k^{(\alpha)}(A)$.

**Taylor.**   About $a > 0$,

$$\log x = \log a + \sum_{k=1}^\infty (-1)^{k+1}\frac{(x - a)^k}{k \, a^k}, \quad |x - a| < a \;\Rightarrow\; \log(A) \approx \log a \, I + \sum_{k=1}^K (-1)^{k+1}\frac{(A - aI)^k}{k \, a^k}, \tag{57}$$

after centering $A$ at $aI$ with $\|A - aI\| < a$.

**Padé.**   For $x = a(1 + u)$, $\log x = \log a + \log(1 + u) \approx \log a + R_{[m/n]}(u)$ with

$$R_{[m/n]}(u) = \frac{p_0 + p_1 u + \cdots + p_m u^m}{1 + q_1 u + \cdots + q_n u^n}, \tag{58}$$

matching the Maclaurin series of $\log(1 + u)$ up to order $m + n$. Matrix form uses $u = (A - aI)/a$:

$$\log(A) \approx \log a \, I + Q_n\left(\tfrac{A - aI}{a}\right)^{-1} P_m\left(\tfrac{A - aI}{a}\right), \tag{59}$$

with $P_m, Q_n$ matrix polynomials.

**Convergence and implementation.**   Each family requires mapping $\sigma(A)$ into its orthogonality domain (e.g., affine map to $[-1, 1]$ for Chebyshev; scaling to enforce $\|A - aI\| < a$ for Taylor/Padé). Coefficients $\{c_k\}$ are computed by quadrature/projection in the scalar domain of the mapped spectrum; evaluation uses only matrix–matrix multiplications via the three–term recurrences. Chebyshev typically yields near–minimax uniform accuracy on $[a, b] \subset (0, \infty)$ and is numerically stable at moderate degrees. Taylor is local; Padé enlarges the effective radius but introduces poles (to be kept outside the target spectrum by choosing $a$ and $[m/n]$ appropriately). Legendre are suitable for weighted $L^2$ targets; Laguerre is natural on $(0, \infty)$ with exponential weights. All forms avoid eigendecomposition and are GPU–friendly.

## A.2 Algorithms for MLN Approximations

**Algorithm 1** Pre-Normalization (shared pre/post for all normalizers)

**Require:** $A \in \mathbb{S}^n_{++}$ (batched or single), tolerance $\varepsilon$
1: $\rho \leftarrow \text{tr}(A)$
2: $\rho \leftarrow \max(\rho, \varepsilon)$                                                          ▷ guard against tiny traces
3: $\tilde{A} \leftarrow A/\rho$
4: **return** $(\tilde{A}, \log \rho)$

**Algorithm 2** Log Taylor Approximation (around $I$; degree $K$)

**Require:** $A \in \mathbb{S}^n_{++}$, truncation $K$, tolerance $\varepsilon$
1: $(\tilde{A}, \log \rho) \leftarrow \text{ScaleByTrace}(A, \varepsilon)$
2: $I \leftarrow I_n, \quad X \leftarrow \tilde{A} - I$
3: $Y \leftarrow 0, \quad X^{(1)} \leftarrow X$
4: **for** $k = 1$ to $K$ **do**
5: $\quad c_k \leftarrow (-1)^{k+1}/k$
6: $\quad Y \leftarrow Y + c_k X^{(k)}$
7: $\quad X^{(k+1)} \leftarrow X^{(k)} X$                                                          ▷ GEMM
8: **end for**
9: **return** $\log(A) \approx (\log \rho) I + Y$

*Cost:* $K$ GEMMs.    *Note:* Ensure $\sigma(\tilde{A} - I) \subset (-1, 1)$ (optionally by spectral scaling or degree increase).

**Algorithm 3** Log Pade Approximation (general $[m/n]$ for $\log(1 + X)$)

**Require:** $A \in \mathbb{S}^n_{++}$, integers $m, n$, tolerance $\varepsilon$
1: $(\tilde{A}, \log \rho) \leftarrow \text{ScaleByTrace}(A, \varepsilon)$
2: $I \leftarrow I_n, \quad X \leftarrow \tilde{A} - I$
3: $(\{p_k\}_{k=0}^m, \{q_k\}_{k=0}^n) \leftarrow$ Padé coeffs for $\log(1 + u)$
4: $P \leftarrow p_0 I, \quad Q \leftarrow q_0 I, \quad X^{(1)} \leftarrow X, \quad K_{\max} \leftarrow \max(m, n)$
5: **for** $k = 1$ to $K_{\max}$ **do**
6: $\quad$ **if** $k \leq m$ **then**
7: $\quad\quad P \leftarrow P + p_k X^{(k)}$
8: $\quad$ **end if**
9: $\quad$ **if** $k \leq n$ **then**
10: $\quad\quad Q \leftarrow Q + q_k X^{(k)}$
11: $\quad$ **end if**
12: $\quad X^{(k+1)} \leftarrow X^{(k)} X$                                                          ▷ GEMM
13: **end for**
14: Solve $QY = P$ for $Y$                                       ▷ use a stable solver (no explicit inverse)
15: **return** $\log(A) \approx (\log \rho) I + Y$

*Cost:* $\max(m, n)$ GEMMs + one linear solve. *Notes:* Cache coeffs; for $[1/1]$ and $[2/2]$ you can use closed forms.

**Algorithm 4** LOG LEGENDRE APPROXIMATION (shifted on $[0, 1]$; degree $K$)

**Require:** $A \in \mathbb{S}^n_{++}$, truncation $K$, tolerance $\varepsilon$
1: $(\tilde{A}, \log \rho) \leftarrow \text{SCALEBYTRACE}(A, \varepsilon)$
2: $I \leftarrow I_n, \quad X \leftarrow 2\tilde{A} - I$                    $\triangleright$ shift to $[0, 1]$ domain
3: $P_0 \leftarrow I, \quad P_1 \leftarrow X$
4: $c_0 \leftarrow -1, \quad c_k \leftarrow (-1)^{k+1} \frac{2k+1}{k(k+1)}$ for $k \geq 1$
5: $Y \leftarrow c_0 P_0 + c_1 P_1$
6: **for** $k = 1$ to $K - 1$ **do**
7:      $P_{k+1} \leftarrow \big((2k+1)X P_k - k P_{k-1}\big)/(k+1)$            $\triangleright$ GEMM
8:      $Y \leftarrow Y + c_{k+1} P_{k+1}$
9: **end for**
10: **return** $\log(A) \approx (\log \rho) I + Y$
*Cost:* $K$ GEMMs. *Note:* Uses closed-form $c_k$ for $\log x$ on $[0, 1]$.

**Algorithm 5** LOG CHEBYSHEV APPROXIMATION ($T_k$ on $[-1, 1]$; degree $K$)

**Require:** $A \in \mathbb{S}^n_{++}$, truncation $K$, bounds $(\lambda_{\min}, \lambda_{\max})$ for $\sigma(\tilde{A})$, tolerance $\varepsilon$
1: $(\tilde{A}, \log \rho) \leftarrow \text{SCALEBYTRACE}(A, \varepsilon)$
2: $I \leftarrow I_n, \quad a \leftarrow (\lambda_{\max} + \lambda_{\min})/2, \quad b \leftarrow \max\{(\lambda_{\max} - \lambda_{\min})/2, \varepsilon\}$
3: $Z \leftarrow (\tilde{A} - aI)/b$                         $\triangleright$ affine map to $[-1, 1]$
4: $T_0 \leftarrow I, \quad T_1 \leftarrow Z$
5: Obtain Chebyshev coefficients $c_k$ for $\log$ on $[-1, 1]$ (precompute or numeric quad)
6: $Y \leftarrow \frac{c_0}{2} T_0 + c_1 T_1$
7: **for** $k = 1$ to $K - 1$ **do**
8:      $T_{k+1} \leftarrow 2Z T_k - T_{k-1}$                     $\triangleright$ GEMM
9:      $Y \leftarrow Y + c_{k+1} T_{k+1}$
10: **end for**
11: **return** $\log(A) \approx (\log \rho) I + Y$
*Cost:* $K$ GEMMs. *Notes:* Tighter $(\lambda_{\min}, \lambda_{\max})$ improves accuracy; $c_k$ can be cached.

**Algorithm 6** LOG LAGUERRE APPROXIMATION ($L_k^{(0)}$ on $[0, \infty)$; degree $K$)

**Require:** $A \in \mathbb{S}^n_{++}$, truncation $K$ (e.g., $K \leq 10$), tolerance $\varepsilon$
1: $(\tilde{A}, \log \rho) \leftarrow \text{SCALEBYTRACE}(A, \varepsilon)$
2: $I \leftarrow I_n, \quad L_0 \leftarrow I, \quad L_1 \leftarrow I - \tilde{A}$
3: Use coefficients for $\log$ projection with $\alpha{=}0$: $c_0 = -\gamma, c_k = -1/k$ for $k \geq 1$
4: $Y \leftarrow c_0 L_0 + c_1 L_1$
5: **for** $k = 1$ to $K - 1$ **do**
6:      $L_{k+1} \leftarrow \big((2k+1)I - \tilde{A}\big) L_k - k L_{k-1}; \quad L_{k+1} \leftarrow L_{k+1}/(k+1)$    $\triangleright$ GEMM
7:      $Y \leftarrow Y + c_{k+1} L_{k+1}$
8: **end for**
9: **return** $\log(A) \approx (\log \rho) I + Y$
*Cost:* $K$ GEMMs. *Notes:* Constants $c_k$ shown for $\alpha{=}0$; other $\alpha$ require re-projection.

**Remark on Backward Pass.** The forward routines above produce $\log(\tilde{A})$ via polynomial or rational forms. The corresponding gradients $\frac{\partial \ell}{\partial \tilde{A}}$ follow from the closed-form sum (Taylor, Padé) or the adjoint of the three-term recurrence (Legendre, Chebyshev, Laguerre) as detailed in Appendix A.3; the final gradient w.r.t. $A$ is obtained via the chain rule in Eq. equation 38 of the main text. All steps are GEMM-only.

## A.3 BACKWARD-PASS DETAILS FOR POLYNOMIAL AND RATIONAL POLYNOMIAL FAMILIES

We derive $\frac{\partial \ell}{\partial \tilde{A}}$ for each family using only matrix multiplications/additions (GEMM). The final gradient w.r.t. the original covariance $A$ follows from the chain rule in Eq. equation 38 of the main text.

**Notation.** For a matrix function $F(\tilde{A})$, write the differential as $dF = \mathcal{J}_F(\tilde{A})[d\tilde{A}]$. We use $\langle X, Y \rangle_F := \mathrm{tr}(X^\top Y)$ and the product rule $d(AXB) = (dA)XB + A(dX)B + AX(dB)$. Throughout, let $U := \frac{\partial \ell}{\partial \log(\tilde{A})}$ denote the upstream gradient.

### A.3.1 TAYLOR (MONOMIAL) EXPANSION

**Forward.**

$$\log(\tilde{A}) \approx \sum_{k=1}^{K} c_k (\tilde{A} - I)^k, \qquad c_k = \frac{(-1)^{k+1}}{k}. \tag{60}$$

**Differential.** For $X(\tilde{A}) := \tilde{A} - I$,

$$d\big[X(\tilde{A})^k\big] = \sum_{j=0}^{k-1} X(\tilde{A})^j (d\tilde{A}) X(\tilde{A})^{k-1-j}. \tag{61}$$

**Adjoint.** Using $\langle U, d\log(\tilde{A})\rangle = \sum_{k=1}^{K} c_k \sum_{j=0}^{k-1} \langle U, X^j (d\tilde{A}) X^{k-1-j}\rangle = \langle \frac{\partial \ell}{\partial \tilde{A}}, d\tilde{A}\rangle$, we obtain

$$\boxed{\frac{\partial \ell}{\partial \tilde{A}} = \sum_{k=1}^{K} c_k \sum_{j=0}^{k-1} (\tilde{A} - I)^{k-1-j} \, U \, (\tilde{A} - I)^{j}.} \tag{62}$$

Convert to $\frac{\partial \ell}{\partial A}$ via Eq. equation 38.

### A.3.2 PADÉ (RATIONAL) APPROXIMANTS

**Forward.**

$$\log(\tilde{A}) \approx R_{[m/n]}(\tilde{A}) = P_m(\tilde{A}) Q_n(\tilde{A})^{-1}. \tag{63}$$

**Differential (Quotient Rule).** Let $Q^{-1} := Q_n(\tilde{A})^{-1}$ (cached in forward pass). Then

$$dR = (dP_m)Q^{-1} - P_m Q^{-1}(dQ_n)Q^{-1}. \tag{64}$$

**Polynomial Differentials.** Write $P_m(\tilde{A}) = \sum_i \Pi_i(\tilde{A})$ and $Q_n(\tilde{A}) = \sum_j \Psi_j(\tilde{A})$ with each term of the form $\Pi_i(\tilde{A}) = A_i^{(L)} \tilde{A}^{p_i} A_i^{(R)}$ (similarly for $\Psi_j$). Then

$$d\Pi_i = \sum_{r=0}^{p_i-1} A_i^{(L)} \tilde{A}^r (d\tilde{A}) \tilde{A}^{p_i-1-r} A_i^{(R)}, \qquad d\Psi_j = \sum_{s=0}^{q_j-1} B_j^{(L)} \tilde{A}^s (d\tilde{A}) \tilde{A}^{q_j-1-s} B_j^{(R)}. \tag{65}$$

**Adjoint.** Using $\langle U, dR \rangle = \langle \frac{\partial \ell}{\partial \tilde{A}}, d\tilde{A}\rangle$ with equation 64–equation 65 gives

$$\boxed{\frac{\partial \ell}{\partial \tilde{A}} = \sum_i \Pi_i'(\tilde{A})^\top Q_n(\tilde{A})^{-\top} U \, \Pi_i(\tilde{A}) \ - \ \sum_j \Psi_j'(\tilde{A})^\top Q_n(\tilde{A})^{-\top} P_m(\tilde{A})^\top U \, Q_n(\tilde{A})^{-\top} \Psi_j(\tilde{A}),}$$

$$\tag{66}$$

where $\Pi_i'(\tilde{A})$ (resp. $\Psi_j'(\tilde{A})$) stacks the left/right polynomial factors appearing with $d\tilde{A}$ in equation 65. Convert to $\frac{\partial \ell}{\partial A}$ via Eq. equation 38.

### A.3.3 Legendre

**Forward Recurrence.** Let $S = 2\tilde{A} - I$ and define

$$(k+1)P_{k+1} = (2k+1)\, S\, P_k - k\, P_{k-1}, \qquad P_0 = I,\ P_1 = S. \tag{67}$$

The expansion is $\log(\tilde{A}) \approx \sum_{k=0}^{K} c_k P_k(S)$.

**Seeds.**

$$\overline{P}_k \ :=\ \frac{\partial \ell}{\partial P_k} \ =\ c_k\, U. \tag{68}$$

**Reverse (Adjoint) Recurrence.** Rewriting equation 67 as $P_{k+1} = a_k S P_k - b_k P_{k-1}$ with $a_k = \frac{2k+1}{k+1}$, $b_k = \frac{k}{k+1}$, the reverse pass satisfies

$$\overline{S} \leftarrow \overline{S} + a_k \overline{P}_{k+1} P_k^\top, \qquad \overline{P}_k \leftarrow \overline{P}_k + a_k S^\top \overline{P}_{k+1}, \qquad \overline{P}_{k-1} \leftarrow \overline{P}_{k-1} - b_k \overline{P}_{k+1}. \tag{69}$$

Initialize $\overline{S} = 0$ and sweep $k = K-1, \ldots, 0$.

**Adjoint to $\tilde{A}$.** Since $S = 2\tilde{A} - I$,

$$\boxed{\frac{\partial \ell}{\partial \tilde{A}} \ =\ 2\,\overline{S} \ =\ 2 \sum_{k=0}^{K-1} a_k \overline{P}_{k+1} P_k^\top.} \tag{70}$$

Convert to $\frac{\partial \ell}{\partial A}$ via Eq. equation 38.

### A.3.4 Chebyshev

**Forward Recurrence.**

$$T_{k+1} \ =\ 2\,\tilde{A}\,T_k \ -\ T_{k-1}, \qquad T_0 = I,\ T_1 = \tilde{A}. \tag{71}$$

The expansion is $\log(\tilde{A}) \approx \sum_{k=0}^{K} c_k T_k(\tilde{A})$.

**Seeds.**

$$\overline{T}_k \ :=\ \frac{\partial \ell}{\partial T_k} \ =\ c_k\, U. \tag{72}$$

**Reverse (Adjoint) Recurrence.** From equation 71,

$$\overline{\tilde{A}} \leftarrow \overline{\tilde{A}} + 2\overline{T}_{k+1} T_k^\top, \qquad \overline{T}_k \leftarrow \overline{T}_k + 2\tilde{A}^\top \overline{T}_{k+1}, \qquad \overline{T}_{k-1} \leftarrow \overline{T}_{k-1} - \overline{T}_{k+1}. \tag{73}$$

Initialize $\overline{\tilde{A}} = 0$ and sweep $k = K-1, \ldots, 0$.

**Adjoint to $\tilde{A}$.**

$$\boxed{\frac{\partial \ell}{\partial \tilde{A}} \ =\ 2 \sum_{k=0}^{K-1} \overline{T}_{k+1} T_k^\top.} \tag{74}$$

Convert to $\frac{\partial \ell}{\partial A}$ via Eq. equation 38.

### A.3.5 Laguerre

**Forward Recurrence.**

$$(k+1)L_{k+1}^{(\alpha)} \ =\ (2k+\alpha+1-\tilde{A})\, L_k^{(\alpha)} \ -\ (k+\alpha)\, L_{k-1}^{(\alpha)}, \qquad L_0^{(\alpha)} = I,\ L_1^{(\alpha)} = (\alpha+1)I - \tilde{A}. \tag{75}$$

Equivalently,

$$L_{k+1}^{(\alpha)} \ =\ a_k L_k^{(\alpha)} \ -\ \frac{1}{k+1} \tilde{A} L_k^{(\alpha)} \ -\ b_k L_{k-1}^{(\alpha)}, \qquad a_k = \frac{2k+\alpha+1}{k+1}, \ \ b_k = \frac{k+\alpha}{k+1}. \tag{76}$$

The expansion is $\log(\tilde{A}) \approx \sum_{k=0}^{K} c_k L_k^{(\alpha)}(\tilde{A})$.

**Seeds.**

$$\overline{L}_k \;:=\; \frac{\partial \ell}{\partial L_k^{(\alpha)}} \;=\; c_k\, U. \tag{77}$$

**Reverse (Adjoint) Recurrence.** From equation 76,

$$\overline{\tilde{A}} \;\leftarrow\; \overline{\tilde{A}} - \frac{1}{k+1}\, \overline{L}_{k+1}\, L_k^{(\alpha)\top}, \qquad \overline{L}_k \;\leftarrow\; \overline{L}_k + a_k\, \overline{L}_{k+1} - \frac{1}{k+1}\, \tilde{A}^\top \overline{L}_{k+1}, \qquad \overline{L}_{k-1} \;\leftarrow\; \overline{L}_{k-1} - b_k\, \overline{L}_{k+1}. \tag{78}$$

Initialize $\overline{\tilde{A}} = 0$ and sweep $k = K-1, \dots, 0$.

**Adjoint to $\tilde{A}$.**

$$\boxed{\frac{\partial \ell}{\partial \tilde{A}} \;=\; -\sum_{k=0}^{K-1} \frac{1}{k+1}\, \overline{L}_{k+1}\, L_k^{(\alpha)\top}.} \tag{79}$$

Convert to $\frac{\partial \ell}{\partial A}$ via Eq. equation 38.

