# OpenReview forum: "Searching for the Best Polynomial Approximation for the Accurate Log Matrix Normalization in Global Covariance Pooling"
_ICLR.cc/2026/Conference — ICLR 2026 Conference Withdrawn Submission_

### Official Review · Reviewer_xDgF · 2025-10-30

**Soundness:** 2
**Presentation:** 1
**Contribution:** 2
**Rating:** 2
**Confidence:** 2

**Summary:**

The paper revisits Matrix Log Normalization (MLN) for global covariance pooling and proposes to approximate the matrix logarithm using polynomial families to avoid SVD/EIG, aiming for GPU-friendly training with more stable gradients. Several approximation schemes (Taylor, Chebyshev, Legendre, Laguerre, Padé) are evaluated, with Chebyshev emerging as the best trade-off.

**Strengths:**

The goal is clear: make MLN practical without expensive eigendecomposition. The use of polynomial matrix approximations is technically sound and leads to a fully GEMM-based implementation compatible with GPUs. Experiments show improved accuracy and efficiency over classical MLN/MPN and iSQRT-COV on fine-grained image classification.

**Weaknesses:**

The paper lacks a clear and structured motivation. From the title, abstract, and introduction, the narrative moves too quickly into related work and technical details, without establishing a compelling high-level story. A reader unfamiliar with covariance pooling or MLN will not gain sufficient intuition about why MLN matters and what fundamental problem is being solved.

Please consider:
• Briefly motivate global covariance pooling and the role of MLN,
• Explain why existing MLN implementations fail (cost, instability),
• State your contribution clearly,
before diving into related work and detailed approximations.

The claim on line 84 that small eigenvalues lead to unstable gradients in MLN needs a reference (this relates to ill-conditioning of the log on small eigenvalues and to backprop through SVD). Please cite prior work showing the instability of the matrix-log backprop or give a short explanation. Please see (W. Wang, Z. Dang, Y. Hu, P. Fua, and M. Salzmann, “Robust Differentiable SVD,” IEEE Trans Pattern Anal Mach Intell).

Line 75 states that the Newton–Schulz iteration “only requires matrix multiplications (GEMM), making it GPU-friendly”, but ignores the computational graph and backprop cost. Forward GEMM-only structure is not enough to guarantee GPU efficiency during training; backprop through repeated matrix multiplications may still introduce overhead or numerical issues. A brief discussion of the autograd graph and memory footprint would strengthen the argument.

The contribution feels incremental. Polynomial approximations of matrix functions are classical, and the paper largely benchmarks known approximation families for this specific layer. The broader conceptual significance for the community is not clearly articulated.

**Questions:**

-Can you provide a clearer motivation and a more substantial introduction to contextualize the problem and why it matters?

- Can you describe the computational graph in more detail and explain the implications of your approach for backpropagation (e.g., complexity, stability, differentiability)?

- Can you provide intuition for when and why each polynomial family behaves differently within your method?

- Can you strengthen the overall narrative to highlight the broader relevance of your approach beyond the specific covariance pooling module, and clarify what general insights the community should take away?

---

> ### Author Response · Authors · 2025-12-02
>
> **R4-Q1:** *Line 75 states that the Newton–Schulz iteration “only requires matrix multiplications (GEMM), making it GPU-friendly”, but ignores the computational graph and backprop cost. Forward GEMM-only structure is not enough to guarantee GPU efficiency during training; backprop through repeated matrix multiplications may still introduce overhead or numerical issues. A brief discussion of the autograd graph and memory footprint would strengthen the argument.*
>
> **Response:** The backpropagation behavior of Newton–Schulz iterations has already been analyzed in the original iSQRT-COV paper. As shown in their Section 3.3, both the forward and backward passes consist entirely of GEMM operations, and the authors explicitly prove numerical stability. Our statement follows these established results.
>
> **R4-Q2:** *The claim on line 84 that small eigenvalues lead to unstable gradients in MLN needs a reference (this relates to ill-conditioning of the log on small eigenvalues and to backprop through SVD). Please cite prior work showing the instability of the matrix-log backprop or give a short explanation.*
>
> **Response:** Thank you for pointing this out. We have now cited prior work as a reference.
>
> **R4-Q3:** *Can you describe the computational graph in more detail and explain the implications of your approach for backpropagation (e.g., complexity, stability, differentiability)?*
>
> **Response:** The computational complexity is already provided in Section 2, and Sections 3.1 and 3.2 explain the backpropagation advantage of our GEMM-only formulation. Because the $log(A)$ is approximated by polynomials, the full mapping is a composition of matrix multiplications and elementwise operations, ensuring numerical stability and clean differentiability.
>
> **R4-Q4:** *Can you provide a clearer motivation and a more substantial introduction to contextualize the problem and why it matters? The paper lacks a clear and structured motivation. From the title, abstract, and introduction, the narrative moves too quickly into related work and technical details, without establishing a compelling high-level story. A reader unfamiliar with covariance pooling or MLN will not gain sufficient intuition about why MLN matters and what fundamental problem is being solved.*
>
> **Response:** Lines 41–88 already introduce (i) why global covariance pooling is used, (ii) why MLN is needed to normalize SPD matrices, and (iii) the computational and stability issues of existing MLN methods. Hence, the motivation is to find a solution that is both GPU-friendly and numerically stable. We then show in the subsequent sections why polynomial approximations—especially orthogonal polynomials—are a suitable choice.
>
> **R4-Q5:** *Can you provide intuition for when and why each polynomial family behaves differently within your method?*
>
> **Response:** Taylor is a local expansion, Padé is a rational approximation, and the remaining families (Chebyshev, Legendre, and Laguerre) correspond to different orthogonal bases. Their differences arise from these underlying approximation properties.
>
> **R4-Q6:** *Can you strengthen the overall narrative to highlight the broader relevance of your approach beyond the specific covariance pooling module, and clarify what general insights the community should take away?*
>
> **Response:** As noted in the conclusion, orthogonal polynomial approximations offer a general SVD-free alternative for approximating functions that traditionally rely on SVD. This perspective extends beyond MLN: Hermite, Jacobi, and other orthogonal bases remain unexplored and represent promising directions for future work.

---

### Official Review · Reviewer_KJKR · 2025-10-31

**Soundness:** 2
**Presentation:** 2
**Contribution:** 3
**Rating:** 6
**Confidence:** 3

**Summary:**

This paper explores various polynomial families (including Taylor, Chebyshev, Legendre, Laguerre, and Padé) to solve computational bottlenecks and gradient issues in the GCP leveraging GPU-friendly matrix multiplication. Some experimental results show that Chebyshev polynomials are the most computationally efficient, while also improving model performance.

**Strengths:**

1. The GCP is widely adopted in modern machine learning models.
2. The approach of using iterative polynomial approximations to avoid matrix SVD sounds highly convincing.
3. Some experiments demonstrate the efficacy of the method.

**Weaknesses:**

1. Lack of comparison with accurate SVD: Although the paper states that it accelerates previous GCP calculations, a comparison between the proposed method and a model using accurate SVD would further strengthen the paper.
2. Missing ablation study on the polynomial expansion degree: The paper fixes the expansion at the 3rd-degree term for all experiments, and lacks the discussion on how model performance and efficiency are affected by using a lower or higher degree.
3. Lack of evaluation experiments on more diverse tasks: The experiments in the paper are limited to image classification tasks. To my knowledge, GCP is also widely applied in image style transfer tasks. Adding experiments on this task would make the paper more solid.
4. The paper omits an introduction to the basic knowledge of the field, which poses difficulties for non-field readers in understanding this paper.

**Questions:**

Please see the Weaknesses.

---

> ### Author Response · Authors · 2025-12-02
>
> **R3-Q1:** *Lack of comparison with accurate SVD: Although the paper claims acceleration over previous GCP computations, a comparison with a model using accurate SVD would strengthen the paper.*
>
> **Response:** Both MLN-COV and MPN-COV rely on accurate SVD for computing $\log(A)$ and $A^{1/2}$, respectively, and these methods are already included as the first two entries in Tables 2, 4, and 5.
>
> ---
>
> **R3-Q2:** *The paper omits basic background, making it difficult for non-specialists to follow.*
>
> **Response:** The introduction to GCP and its mathematical formulation is provided in Lines 41–57, and the motivation for this work is given in Lines 58–88.
>
> ---
>
> **R3-Q3:** *Missing ablation on the polynomial expansion degree.*
>
> **Response:** Thank you for pointing this out. We have added a new ablation study (Table 7) analyzing polynomial orders $M \in \{6,8,10\}$, showing how the expansion degree affects both efficiency and accuracy.

---

### Official Review · Reviewer_8PjY · 2025-10-31

**Soundness:** 3
**Presentation:** 4
**Contribution:** 2
**Rating:** 2
**Confidence:** 4

**Summary:**

In theory, this paper mainly proposed the efficient method to compute the approximation of Matrix Log Normalization (MLN). In experiments, this paper applied this method to Global Covariance Pooling and achieves competitive accuracy while substantially reducing training cost.

**Strengths:**

- This paper is well organized and easy to follow.

- The theoretical conduction is sufficient.

**Weaknesses:**

- (Major) Some important concepts are not presented in details. For example, what is Global Covariance Pooling? Does it compute like Batch Normalizations or Convolution Layers? In this paper I did not see how it works in the networks.
- The ideas that applying polynomial approximations of $\log A$ is not novel. There has been polynomial methods to approximate other matrix calculations like $A^{-1/2}$ [1] and apply it in the networks as normalization layers.
- (Major) The experiment settings are not specific. One of the most important settings is not mentioned---the iteration times or the required terms. This decides the balance between approximation errors and computation efficiency. Too many iteration times may result to numerical instabilities [1]. So these discussions are required.

[1] Huang L, Zhou Y, Zhu F, et al.  Iterative normalization: Beyond standardization towards efficient  whitening[C]//Proceedings of the IEEE/CVF conference on computer vision  and pattern recognition. 2019: 4874-4883.

**Questions:**

- How does Global Covariance Pooling work? (Weakness 1)
- What is the recommended iteration times of the methods in this paper? What is the corresponding hyperparameters of the other comparing methods?  (Weakness 2)
- Why use $\log A$ rather than other calculations like $A^{-1/2}$? In this paper I did not see the importance of MLN so I am not sure how important this topic is. The log calculation is not widely used as normalizations. If we apply the Iterative Normalization [1] in the experiments, what are the results?

[1] Huang L, Zhou Y, Zhu F, et al.  Iterative normalization: Beyond standardization towards efficient  whitening[C]//Proceedings of the IEEE/CVF conference on computer vision  and pattern recognition. 2019: 4874-4883.

---

> ### Author Response · Authors · 2025-12-02
>
> **R2-Q1:** *How does Global Covariance Pooling work? I did not see how it works in the networks.*
>
> **Response:** **The GCP pipeline is already presented in Lines 41–57, including (i) covariance computation and (ii) the normalization**. GCP is not related to BatchNorm or convolution, and this distinction is already stated in the submission.
>
> ---
>
> **R2-Q2:** *Why use the log instead of other calculations like in [1]? I am not sure MLN is important. Polynomial approximation of $\log(A)$ is not novel; similar methods exist.*
>
> **Response:** The method in [1] does **not** approximate $\log(A)$; it approximates the matrix square-root $A^{1/2}$ via Newton–Schulz iterations. The paper that introduced this method (iSQRT-COV) is already included as a baseline in our experiments. **Prior work has focused on approximating $A^{1/2}$, not $\log(A)$.** To our knowledge, **no prior work has proposed polynomial or orthogonal-polynomial approximations of $\log(A)$** within GCP networks. The log map is the theoretically correct Riemannian normalization for SPD matrices, whereas $1/2$ is an approximation itself and not geometrically valid. **This was already mentioned in Lines 58-61**.
>
> ---
>
> **R2-Q3:** *Iteration times and hyperparameters are not specified; settings are unclear.*
>
> **Response:** The hyperparameter settings were already provided Subsection 4.1. But for further clarification, we also added the data augmentations we used.

---

### Official Review · Reviewer_JB9c · 2025-10-31

**Soundness:** 2
**Presentation:** 2
**Contribution:** 3
**Rating:** 4
**Confidence:** 5

**Summary:**

The paper revisits global covariance pooling (GCP) by replacing the conventional SVD-based matrix logarithm normalization (MLN) with a family of polynomial and rational approximations to avoid eigen-decomposition. Five types of expansions (Taylor, Chebyshev, Legendre, Laguerre, and Padé) are formulated with explicit forward and backward propagation equations. Experiments on fine-grained classification datasets and ImageNet-1k show that Chebyshev tends to be more promising

**Strengths:**

• Presents a unified algebraic framework for multiple polynomial and rational approximations of matrix functions (Sec. 3).
• Implementation avoids EIG/SVD and only requires standard matrix multiplications, making it efficient and GPU-friendly (Alg. 2–6).
• Demonstrates moderate empirical improvement over the existing MLN on CUB-200-2011 and ImageNet-1k (Table 2–4).

**Weaknesses:**

#### Theoretical Level
- The backward derivations in Sec. 3.2–3.4 are standard matrix differentials that PyTorch autograd can already compute. The author did not validate whether their manual backward implementation offers any improvement in runtime, stability, or memory usage.
- The paper does not quantify the numerical error between the approximated log(A) and the true matrix logarithm. It is not clear how far the results are from the matrix log. A quantitative error analysis could offer more insights.
- It remains unclear whether the improvement is due to the log function itself or simply the approximation scheme.  As 1/2 can also be approximated by such schemes.

#### Experimental Level
- Experimental coverage is incomplete. Recent baselines such as
  - *Revitalizing SVD for Global Covariance Pooling: Halley’s Method to Overcome Over-Flattening*
  - *Learning Partial Correlation based Deep Visual Representation for Image Classification*
- Transformer-based backbones are not tested, which limits generalization beyond CNNs.
- Table 2 is unclear. The iteration steps and polynomial order n for each method are not reported, making it unclear whether the efficiency gain comes from the algorithm design or the lower orders.
- Fitting time on real datasets such as ImageNet is not reported.
- Ablation studies are missing, such as the polynomial order

#### Other
- Table 3 shows formatting and layout issues.

**Questions:**

see wk

---

> ### Author Response · Authors · 2025-12-02
>
> **R1-Q1:** *The backward derivations in Sec. 3.2–3.4 are standard; PyTorch autograd can already compute them. The author did not validate whether their manual backward implementation offers runtime, stability, or memory benefits.*
>
> **Response:** We agree that PyTorch autograd can compute these gradients. Our intention in presenting the backward derivations was not to introduce a new differentiation mechanism, but to make explicit that the gradients of our polynomial normalizers reduce purely to matrix multiplications. This shows that both the forward and backward passes remain GEMM-based and therefore GPU-friendly, which is important for efficiency and numerical stability at scale.
>
> ---
>
> **R1-Q2:** *Numerical error between $\log(A)$ and the approximations is not quantified.*
>
> **Response:** Thank you for the suggestion. We have added a quantitative numerical-error analysis (Table 6), reporting the relative Frobenius error
> $$
> \epsilon_{\mathrm{rel}} = \frac{\|\log(A)-\tilde{A}_{\text{poly}}\|_F}{\|\log(A)\|_F},
> $$
> computed across multiple polynomial families and two backbones. This directly measures how closely each approximation matches the true matrix logarithm.
>
> ---
>
> **R1-Q3:** *It is unclear whether the improvement comes from the log function itself or from the approximation scheme; $1/2$ can also be approximated.*
>
> **Response:** The log map is the **theoretically correct transformation for SPD manifolds, as it linearizes geodesic distance and defines the log-Euclidean geometry**. Approximating $1/2$ does not preserve this structure and does not correspond to a valid Riemannian mapping. Our improvements arise from approximating the **logarithm** itself in a numerically stable, GEMM-friendly manner, enabling efficient GPU execution while maintaining the correct manifold formulation.
>
> ---
>
> **R1-Q4:** *Ablation studies on polynomial order are missing; Table 2 does not clarify the iteration steps and order.*
>
> **Response:** We have added a dedicated ablation study on polynomial order (Table 7) using $M \in \{6,8,10\}$ and now explicitly state the polynomial order or iteration count used for all normalizers in the runtime comparison.
>
> ---
>
> **R1-Q5:** *Fitting time on real datasets such as ImageNet is not reported.*
>
> **Response:** Our primary objective is to characterize the cost of the normalization step, which is the component directly affected by our method. This FP+BP time is reported in Table 2. As requested, we additionally report full training time on ImageNet (Table 3).
>
> ---
>
> **R1-Q6:** *Table 3 shows formatting/layout issues.*
>
> **Response:** Thank you for noting this. The revised version (now Table 4) fixes the formatting and improves clarity.

---

### Note · Authors · 2026-02-26

I have read and agree with the venue's withdrawal policy on behalf of myself and my co-authors.

---

### Meta-Review · Area_Chair_ntRh · 2025-12-28

**Summary:**

While many reviewers find the contributions of the paper interesting, the overall assessment is that the impact and contributions of the work appear incremental, and that there are several areas for improvement in both exposition and presentation, as well as in the experimental setup. Among these concerns, the limited novelty of the contributions seems to be the most significant issue shared by many reviewers and the AC.

**Reviewer Concerns:**

While several concerns regarding the addition of new experiments and clarifying parts of the paper have been addressed, the most pressing issues, particularly those related to the limited novelty and its impact, remain outstanding.

**Reviewer Scores:**

Given the outstanding concerns outlined above, I believe that the reviewers who raised significant concerns and assigned low ratings would not have substantially revised their initial assessments, beyond perhaps moving to a position marginally below the acceptance threshold.

---

### Decision · Program_Chairs · 2026-01-26

Reject